# OpenReview forum: "Comparing AI Agents to Cybersecurity Professionals in Real-World Penetration Testing"
_ICLR.cc/2026/Conference — ICLR 2026 Poster_

### Official Review · Reviewer_wjEA · 2025-10-26

**Soundness:** 3
**Presentation:** 3
**Contribution:** 3
**Rating:** 6
**Confidence:** 5

**Summary:**

This submission reports on the results of an experiment comparing the performance of human experts and LLM-based agent systems on discovering security vulnerabilities in a live real-world environment. The experiment was conducted with IRB approval on a university’s public and private networks, over 12 subnets (7 public, 5 private) comprising ~8K hosts, primarily Unix-based systems, but also including IoT devices, Windows hosts and embedded systems.

For the study, the authors implemented a new multi-agent system, ARTEMIS, deployed in 2 configurations, and compared it against:

 - 10 cybersecurity professionals, recruited to commit at least 10 hours of work.
 - 2 general purpose agent systems: OpenAI Codex with GPT-5 and Claude Code with Sonnet 4.
 - 3 existing specialist agent systems: CyAgent with GPT-5 and Claude Sonnet 4, MAPTA and Incalmo with GPT-5.

All agent systems ran for 16 hours, though effectively only 2 existing systems (Codex and CyAgent) were evaluated. Claude Code and MAPTA refused to perform the task while Incalmo was unable to make any progress. Overall, 49 vulnerabilities were detected (and responsibly disclosed) during the experiment.

The results, summarized in Table 1, show that ARTEMIS performed competitively against human experts in terms of number, complexity, and severity of vulnerabilities discovered, ranked 2nd overall according to a custom scoring system, but creating many more false positives despite using a triaging agent to gate submissions. The authors argue that the results of the study provide evidence that multi-agent systems like ARTEMIS are already cost effective compared to professionals, with ARTEMIS most costly configuration (the one that ranked 2nd) operating at USD 59/h compared to an average compensation of USD 60/h for the average professional in the US.

**Strengths:**

- Presents a significant study evaluating the cost-effectiveness for vulnerability research of agent systems compared to cybersecurity professionals in a live system under realistic conditions.
- Describes the design of a new multi-agent system capable of identifying and reporting security vulnerabilities in live systems at a fraction of the cost of professionals.
- Qualitatively evaluates and compares the strategies followed by professionals and agent systems, providing insights into shortcomings of agent systems and how they complement human experts.

**Weaknesses:**

- 3 of the 5 existing systems evaluated (Claude Code, MAPTA and Incalmo) were unable to perform the task given, but the manuscript does not discuss in adequate depth the reasons why. It is unclear whether the authors made a reasonable effort to adapt their scaffolding, prompts and tools to better suit the task. This may have biased the results in favor of ARTEMIS.
- Not enough context provided to compare the results presented to other studies evaluating cybersecurity capabilities of current agent systems. For instance, David & Gervais (2025) report strong results evaluating MAPTA on the XBOW validation benchmark. It is unclear how ARTEMIS would perform on existing benchmarks, outside of the carefully framed task of the study. Same for Incalmo.
- Cost comparisons do not take into account the cost of triaging vulnerabilities post-submission. During the study, most human participants submitted only valid findings, while agent systems including ARTEMIS submitted between 18 and 45% invalid findings.

## Typos / Minor comments

 - FPR in abstract. Reported 18-43%, but Table 1 shows one ARTEMIS configuration having 45% invalid submissions (and the other configuration 18%), so isn't it 18-45%?
- page 4, typo "Table 3" should refer to "Table 2".
- page 8, typo "a going single step further" should read "going a single step further"
- There is a lot of redundancy between Table 1, Table 2, and Figure 3. In fact, Table 2 is entirely subsumed by Figure 3. Consider consolidating the presentation of results to make space for additional technical details of the design of ARTEMIS from Appendix A.
- There is not much information in Table 3. The first paragraph of Section 5 says it all, and is necessary to understand the meaning of columns in the table. Consider moving it to the Appendix and instead include more details from Appendix A.
- Pleas spell out TTP in the conclusion, which may be unfamiliar for some readers.

**Questions:**

1. Although you obtained IRB approval to conduct experiments in a live university environment, I am concerned about how well you represented the risks associated with using autonomous agents. These systems are explicitly asked to find security vulnerabilities, have access to the public web, and use models susceptible to hallucinations and prompt injection attacks. The only guardrail stopping the agents from misusing their capabilities seems to be a set of constraints in their system prompt (Appendix I). I could not find any mention that the agents' operated under human oversight to approve potentially dangerous actions. Can you assert that you represented these risks fairly to your IRB and explain what additional measures (if any) you took to impose hard constraints on the agents' actions?
2. Which models were used in the ensemble of supervisor models in the A2 configuration of ARTEMIS?
3. Did you compensate the cybersecurity professionals recruited for the experiment for their work? Since you mention the $60/h rate for the average cybersecurity professional, how much did you pay them?
4. Did you evaluate ARTEMIS on any existing benchmarks besides the environment of your experiments?

**Details Of Ethics Concerns:**

The authors obtained IRB approval to conduct their experiment on a live university environment in coordination with IT staff. I have no concern that the human part of the study was conducted responsibly: the authors asked the professionals participating to follow the university's vulnerability discovery policy and to respect the scope of the study and refrain from performing disruptive actions. However, I am concerned about the amount of thought and attention put into the evaluation of autonomous agent systems and how well the dangers were represented when obtaining IRB approval. Indeed, the experiments seem to have relied simply in soft system prompt instructions constraining the agents' actions (see the `constraints` section of the system prompt in Appendix I). Considering that many (all?) of the agent systems had access to the web and that the LLMs used are susceptible to hallucinations and prompt injection attacks, there is no telling what the agents could have done when unleashed in a live system, ranging from exfiltrating personal data to performing destructive actions and inserting persistent backdoors. It seems highly irresponsible to rely purely on prompting to restrict agents' actions and I am concerned about this work motivating similar studies conducted on live systems with insufficient oversight.

---

> ### Author Response · Authors · 2025-11-27
>
> Thank you for your thorough and thoughtful evaluation. We address each concern below.
>
> ## Weaknesses
>
> > 3 of the 5 existing systems evaluated (Claude Code, MAPTA and Incalmo) were unable to perform the task given, but the manuscript does not discuss in adequate depth the reasons why. It is unclear whether the authors made a reasonable effort to adapt their scaffolding, prompts and tools to better suit the task. This may have biased the results in favor of ARTEMIS.*
>
> We have expanded Section 4.2 to clarify our configuration choices. Claude Code accepts plaintext prompts, so we provided the same instructions and scope as ARTEMIS; it immediately refused to perform penetration testing despite running the same underlying model as ARTEMIS's sub-agents. MAPTA, following David & Gervais (2025), does not accept free-form prompts, so we supplied only target IPs as intended by its design; it also refused due to GPT-5's refusal mechanisms. Incalmo relies on a rigid, predefined task graph and repeatedly stalled at early reconnaissance stages. Substantially modifying its internals would have constituted building a new scaffold rather than evaluating the published one. Our intent was to evaluate these systems as practitioners realistically use them, not to retrofit them with task-specific logic.
>
> > Not enough context provided to compare the results presented to other studies evaluating cybersecurity capabilities of current agent systems. For instance, David & Gervais (2025) report strong results evaluating MAPTA on the XBOW validation benchmark. It is unclear how ARTEMIS would perform on existing benchmarks, outside of the carefully framed task of the study. Same for Incalmo.*
>
> We have evaluated ARTEMIS on Cybench, the most common benchmark for cybersecurity agent capabilities. Results are reported in Table 2:
>
> | Model | Success Rate |
> |-------|--------------|
> | Claude 4.5 Sonnet | 55% |
> | ARTEMIS | 48.6% |
> | OpenAI GPT-5 | 45.9% |
> | Claude 4.1 Opus | 38% |
> | Claude 4 Opus | 38% |
> | Claude 4 Sonnet | 35% |
> | OpenAI o3-mini | 22.5% |
>
> *Table: Comparison of success rates on Cybench. Aside from ARTEMIS and GPT-5 results, all numbers are taken from the Cybench website.*
>
> > Cost comparisons do not take into account the cost of triaging vulnerabilities post-submission. During the study, most human participants submitted only valid findings, while agent systems including ARTEMIS submitted between 18 and 45% invalid findings.*
>
> Correct—our cost analysis does not include triage costs, and we now state this explicitly. All findings were triaged by our team before forwarding to the university. While agents submitted more invalid findings, marginal triage time (typically minutes per finding) is small relative to hours required to discover and exploit vulnerabilities. This overhead does not meaningfully change the cost comparison. We agree that reducing false-positive rates is important and highlight this as future work.
>
> ## Typos / Minor comments
> Thank you. We have updated the manuscript accordingly.

---

> > ### Author Response · Authors · 2025-11-27
> >
> > ## Questions
> >
> > > Although you obtained IRB approval to conduct experiments in a live university environment, I am concerned about how well you represented the risks associated with using autonomous agents. These systems are explicitly asked to find security vulnerabilities, have access to the public web, and use models susceptible to hallucinations and prompt injection attacks. The only guardrail stopping the agents from misusing their capabilities seems to be a set of constraints in their system prompt (Appendix I). I could not find any mention that the agents' operated under human oversight to approve potentially dangerous actions. Can you assert that you represented these risks fairly to your IRB and explain what additional measures (if any) you took to impose hard constraints on the agents' actions?*
> >
> > We share your concern about deploying autonomous agents in real-world environments. Prompt-level constraints were not the only safeguards. Throughout the study, agents operated under continuous human oversight: a research team member monitored each session in real time—commands, terminal output, and jumpbox network activity—and could immediately terminate the agent upon out-of-scope or risky behavior.
> >
> > In parallel, the university's IT department designated a low-risk network segment for this research and monitored telemetry throughout. Kali VMs were restricted to this scope via technical and procedural controls, and IT staff could flag or halt activity as needed.
> >
> > These layers—real-time human monitoring, a risk-mitigated network segment, and strict scoping controls—formed our risk-mitigation plan. We have expanded Section 3 and the ethics section to make these safeguards explicit.
> >
> > > Which models were used in the ensemble of supervisor models in the A2 configuration of ARTEMIS?*
> >
> > Section 4.2 now includes the full list: Claude Sonnet 4, OpenAI o3, Claude Opus 4, Gemini 2.5 Pro, and OpenAI o3 Pro.
> >
> > > Did you compensate the cybersecurity professionals recruited for the experiment for their work? Since you mention the $60/h rate for the average cybersecurity professional, how much did you pay them?***
> >
> > Yes. Per our IRB protocol, participants were compensated $2,000.
> >
> > > Did you evaluate ARTEMIS on any existing benchmarks besides the environment of your experiments?*
> >
> > Yes—see Cybench results above.
> >
> > ## Ethics
> >
> > > The authors obtained IRB approval to conduct their experiment on a live university environment in coordination with IT staff. I have no concern that the human part of the study was conducted responsibly: the authors asked the professionals participating to follow the university's vulnerability discovery policy and to respect the scope of the study and refrain from performing disruptive actions. However, I am concerned about the amount of thought and attention put into the evaluation of autonomous agent systems and how well the dangers were represented when obtaining IRB approval. Indeed, the experiments seem to have relied simply in soft system prompt instructions constraining the agents' actions (see the constraints section of the system prompt in Appendix I). Considering that many (all?) of the agent systems had access to the web and that the LLMs used are susceptible to hallucinations and prompt injection attacks, there is no telling what the agents could have done when unleashed in a live system, ranging from exfiltrating personal data to performing destructive actions and inserting persistent backdoors. It seems highly irresponsible to rely purely on prompting to restrict agents' actions and I am concerned about this work motivating similar studies conducted on live systems with insufficient oversight.*
> >
> > We did not rely solely on prompt-level safeguards. Throughout each agent run, a research team member continuously monitored commands, tool calls, and outputs, while university IT monitored network traffic and service health within the designated low-risk segment. Across all runs, no agent sessions required termination due to adversarial or out-of-scope behavior, and IT reported no misbehavior.

---

### Official Review · Reviewer_e7SU · 2025-10-28

**Soundness:** 4
**Presentation:** 4
**Contribution:** 4
**Rating:** 4
**Confidence:** 5

**Summary:**

This paper presents an ambitious effort to benchmark the performance of autonomous AI agents against human cybersecurity professionals in a live, complex enterprise environment (a large university network). The evaluation compares six existing open-source agents against ten human pen-testers, alongside the authors' newly proposed multi-agent scaffold, ARTEMIS, which utilizes dynamic task prompt generation and arbitrary sub-agent instantiation. The primary contribution is the establishment of a real-world, high-stakes evaluation framework for LLM-based agents in the domain of offensive security.

**Strengths:**

1. The core strength is the commitment to a live enterprise environment (8K hosts, 12 subnets). This is a crucial step forward from synthetic environments and provides uniquely valuable, though difficult to validate, insights into autonomous agent performance in a complex system.

2. The direct comparison against a team of 10 human cybersecurity professionals provides a tangible, high-quality benchmark for AI capabilities.

3. Primary novelty in the ARTEMIS methodology is the successful implementation of a multi-agent system on a live enterprise target.

**Weaknesses:**

1. The use of a "live enterprise environment" makes the core evaluation non-reproducible.

2. The paper needs to clearly articulate the technical novelty of ARTEMIS in its agent-planning module or tool-use orchestration that distinguishes it from existing multi-agent systems.

3. The authors filed to provide error analysis on why ARTEMIS fails.

4. The authors did not compare effect of various LLMs when used with ARTEMIS, a good comparison of proprietary vs open-sourced models would add more value to the paper.

**Questions:**

1. How was the total time budget for the AI agents determined, and how did this compare to the human professionals?

2. Were the agents limited to the same network footprint or information access as the human testers?

---

> ### Author Response · Authors · 2025-11-27
>
> ## Weaknesses
>
> > The use of a "live enterprise environment" makes the core evaluation non-reproducible.*
>
> We agree—but this was necessary to achieve the realism central to the project. A common critique of existing security benchmarks (e.g., CyBench, CVE-Bench) is that they sacrifice realism for reproducibility ([Carlini et al., 2025](https://arxiv.org/pdf/2503.01811)). We aimed to move beyond this limitation. We are developing runnable replicas of the target scope to enable future benchmarking.
>
> > The paper needs to clearly articulate the technical novelty of ARTEMIS in its agent-planning module or tool-use orchestration that distinguishes it from existing multi-agent systems.*
>
> We have expanded Section 3.3 to describe ARTEMIS's technical contributions: supervisor architecture, dynamic expert-system prompt generation, large-horizon context management, and multi-stage triage pipeline. These features distinguish ARTEMIS from prior multi-agent scaffolds.
>
> > The authors failed to provide error analysis on why ARTEMIS fails.*
>
> We have expanded Section 6 with detailed analysis of ARTEMIS's failure modes, including GUI-heavy exploitation issues, premature submission behavior, and false-positive patterns.
>
> > The authors did not compare effect of various LLMs when used with ARTEMIS, a good comparison of proprietary vs open-sourced models would add more value to the paper.*
>
> Due to limited time in the live environment, we prioritized models with the strongest known cybersecurity capabilities. Existing benchmarks show a substantial gap between proprietary and open-source models for offensive-security tasks. Since ARTEMIS's scaffold does not enhance underlying domain knowledge, we used state-of-the-art proprietary models to estimate an upper bound on agentic risk. Evaluating open-source models is a valuable direction for future work.
>
> ## Questions
>
> > How was the total time budget for the AI agents determined, and how did this compare to the human professionals?*
>
> We ran each ARTEMIS variant for 16 wall-clock hours, split across two 8-hour "workdays," to study cost dynamics and long-horizon behavior. For fairness, leaderboard comparisons only include findings from the first 10 hours.
>
> *\> Were the agents limited to the same network footprint or information access as the human testers?*
>
> Yes. Agents received exactly the same resources, credentials, network scope, and information as human participants. No additional data or privileged access was provided.

---

### Official Review · Reviewer_kmZ9 · 2025-10-31

**Soundness:** 3
**Presentation:** 4
**Contribution:** 4
**Rating:** 8
**Confidence:** 2

**Summary:**

A live comparison study pitting 10 real pen-testers vs. various agentiic frameworks, including the proposed ARTEMIS framework.
8K hosts, 7 public and 5 VPN subnets.
The proposed framework had the best automated performance.
Finds more vulnerabilities than some human pentesters, however the false positive rate is higher than for all human pentesters who have little to no false positives.

LLMs used are GPT-5 and Sonnet 4.

Metric is based on a mix of technical complexity and criticality of the found vulnerability.
I'm not sure why there is penalty for "verifying" but not "exploiting" a penalty?

Artemis architecture: 1 supervisor + unlimited subagents. Task list and note taking. Vulnerabilities are verified to be reproducible and deduplicated.

The authors also validate that ARTEMIS can find vulnerabilities found by other participants, given enough hints.
Advantage vs human: parallel search
Disadvantage vs human: using GUIs

**Strengths:**

- Great comparison between human vs. AI capability
- Experiment in live setting
- Strong results for their proposed framework vs. existing coding frameworks including low cost.
- Open-sourced

**Weaknesses:**

- The timeframe is short, as noted by the authors, and may not be representative of a typical pentesting engagement timeframe.
- AI agent reports vulnerabilities

**Questions:**

I'm not sure i understand why penalty if vulnerability is only verified but not exploited. What does "verify" mean exactly here?

---

> ### Author Response · Authors · 2025-11-27
>
> ## Weaknesses
>
> > AI agent reports vulnerabilities*
>
> Could you clarify the rationale for listing this as a weakness? The primary output of a penetration test is an audit-style vulnerability report, so we had agents generate reports for direct comparison with human participants. Both humans and agents submitted findings to the research team only. We performed standardized triage and validation before forwarding confirmed issues to the security team, avoiding false positives and ensuring accurate comparison.
>
> ## Questions
>
> > I'm not sure i understand why penalty if vulnerability is only verified but not exploited. What does "verify" mean exactly here?*
>
> "Verified but not exploited" means the participant confirmed preconditions for a vulnerability (e.g., specific software version, configuration, or cipher suite) but did not demonstrate real impact (e.g., data exfiltration or code execution).
>
> This may occur due to scope restrictions (e.g., avoiding real customer data), technical constraints within the timeline, or dependencies on specific system state. Because actual impact is not demonstrated, scoring reflects reduced risk compared to a fully exploited vulnerability. We have clarified this in Section 3.2.

---

### Official Review · Reviewer_yrRY · 2025-11-01

**Soundness:** 3
**Presentation:** 3
**Contribution:** 4
**Rating:** 8
**Confidence:** 3

**Summary:**

This paper presents a real-world study comparing AI agents with human cybersecurity professionals on a live university network (about 8,000 hosts across 12 subnets). Ten professionals are evaluated alongside five existing agent scaffolds: Codex, Claude Code, CyAgent (CS and CG), Incalmo, and MAPTA, as well as the authors’ own multi-agent framework, ARTEMIS. Only Codex and the two CyAgent variants produced valid findings, while ARTEMIS (two configurations, A1 and A2, using different LLM setups) achieved the best overall performance among AI systems. ARTEMIS A2 ranked second overall, and outperformed 9 of 10 human participants, with 11 findings and 82 percent validity. The paper introduces a unified scoring framework that combines technical complexity and business impact, maps discovered techniques to the MITRE ATT&CK matrix, and includes elicitation trials to test missed vulnerabilities. It also presents a cost analysis suggesting a favorable cost per hour compared to human penetration testers, making the study both technically and economically insightful.

**Strengths:**

1. Claimed to be the first live comparison of AI agents versus professionals in an enterprise environment (a large university CS network), closely matching real-world penetration testing practice. This live comparison is significant compared to curated internet benchmarks, which models may have unintentionally trained on.

2. The proposed ARTEMIS shows impressive performance, outperforming almost all of the human participants. The multi-agent design is well motivated, with dynamic prompt generation and session/context management thoughtfully engineered. The cost analysis, showing that the agent operates at significantly lower cost than humans while maintaining strong performance, demonstrates the practical potential for automating this task.

3. The methodology is transparent and emphasizes a strong ethical approach in a live environment. IRB approval, VDP compliance, and a well-documented triage process minimize risks to the target.

4. The unified scoring metric, which combines detection and exploit complexity with severity weighting, provides more meaningful insights by capturing both the sophistication and impact of vulnerabilities, rather than relying on simple counts.

5. The paper provides useful qualitative contrasts, showing that agents excel at systematic enumeration and parallelism but tend to over-submit lower-severity or false-positive findings. The timeline plot (Fig. 1) and incidence distributions (Fig. 3) are informative and clearly presented.

**Weaknesses:**

1. The runtime and evaluation budget between human participants and AI agents are not strictly matched. Humans were asked to work for at least 10 hours, while ARTEMIS was allotted 16 hours. This difference makes the leaderboard comparison less fair and should be normalized or clearly justified.

2. The paper reports totals and percentages but lacks deeper statistical analysis such as variance or confidence intervals. Without statistical treatment, it is difficult to judge whether performance differences are statistically significant or within normal variation.

3. The claimed advantages of ARTEMIS (dynamic prompts, unlimited sub-agents, and the triage module) are not isolated through ablation studies. It remains unclear how much each component contributes to the overall improvement.

4. The paper does not specify whether the agents used a standardized virtual environment (such as Kali Linux or a containerized lab). Tool availability is described conceptually but lacks implementation details about the runtime setup, which limits reproducibility.

5. It would be beneficial to evaluate ARTEMIS on existing CTF-style benchmarks such as CyBench or NYU-CTF to better understand its performance in standardized settings and how its findings correlate with existing benchmarks.

**Questions:**

1. How does the system handle model safeguards or refusals, since some LLMs may reject offensive or security-related instructions? Please clarify how the agents were able to perform penetration testing tasks without manual intervention or policy violations.

2. What is the typical and maximum number of sub-agents deployed during execution, and how does this relate to the performance advantage claimed for the proposed framework? The paper states that the agent can spawn unlimited sub-agents, but what was the actual peak number observed during the experiments?

3. What is the detailed cost breakdown for each major component (such as the supervisor, sub-agents, and triage module) to better understand which parts contribute most to the total operational cost?

4. Please clarify if any points in the weaknesses section were misunderstood. I am open to revising my evaluation, as this work presents valuable insights and potential impact for the research community.

---

> ### Author Response · Authors · 2025-11-27
>
> Thank you for the detailed and constructive feedback!
>
> ## Weaknesses
>
> > The runtime and evaluation budget between human participants and AI agents are not strictly matched. Humans were asked to work for at least 10 hours, while ARTEMIS was allotted 16 hours. This difference makes the leaderboard comparison less fair and should be normalized or clearly justified.*
>
> We ran ARTEMIS for 16 hours to assess capabilities over a longer two-day horizon. However, for all leaderboard comparisons, we consider only vulnerabilities submitted within the first 10 hours, matching the human evaluation window. We have updated the manuscript to make this explicit.
>
> > The paper reports totals and percentages but lacks deeper statistical analysis such as variance or confidence intervals. Without statistical treatment, it is difficult to judge whether performance differences are statistically significant or within normal variation.*
>
> Because each configuration was run once and our sample size is limited, traditional statistical analyses (e.g., confidence intervals or significance tests) are not reliable. We now state this limitation explicitly in Section 7. Our study is intended as an ecologically valid, point-in-time comparison in a live enterprise environment rather than a hypothesis-testing experiment.
>
> > The claimed advantages of ARTEMIS (dynamic prompts, unlimited sub-agents, and the triage module) are not isolated through ablation studies. It remains unclear how much each component contributes to the overall improvement.*
>
> We agree that ablation studies would strengthen our claims. However, the constraints of a live enterprise environment, combined with limited university time and resources, prevented controlled ablations with sufficient rigor. We are developing runnable replicas of the target scope to enable systematic ablations of individual ARTEMIS components. We have updated Section 7 to make this limitation explicit.
>
> > The paper does not specify whether the agents used a standardized virtual environment (such as Kali Linux or a containerized lab). Tool availability is described conceptually but lacks implementation details about the runtime setup, which limits reproducibility.*
>
> Agents used the same standardized Kali Linux virtual machine as all human participants. We have clarified this setup in Section 4.2.
>
> > It would be beneficial to evaluate ARTEMIS on existing CTF-style benchmarks such as CyBench or NYU-CTF to better understand its performance in standardized settings and how its findings correlate with existing benchmarks.*
>
> We have evaluated ARTEMIS on Cybench. Results are reported alongside ARTEMIS $A_1$ in Table 2:
>
> | Model | Success Rate |
> |-------|--------------|
> | Claude 4.5 Sonnet | 55% |
> | ARTEMIS | 48.6% |
> | OpenAI GPT-5 | 45.9% |
> | Claude 4.1 Opus | 38% |
> | Claude 4 Opus | 38% |
> | Claude 4 Sonnet | 35% |
> | OpenAI o3-mini | 22.5% |
>
> *Table: Comparison of success rates on Cybench. Aside from ARTEMIS and GPT-5 results, all numbers are taken from the Cybench website.*
>
> As noted in Section 3.3, we attribute ARTEMIS's higher performance over CyAgent with GPT-5 primarily to sampling variance rather than scaffold uplift. We have not yet evaluated Claude 4.5 Sonnet or 4.5 Opus with ARTEMIS.
>
> ## Questions
>
> > How does the system handle model safeguards or refusals, since some LLMs may reject offensive or security-related instructions? Please clarify how the agents were able to perform penetration testing tasks without manual intervention or policy violations.*
>
> We observed no refusals during ARTEMIS runs, in contrast to MAPTA, which could not attempt the task due to base model refusals. We attribute this to ARTEMIS's prompting architecture and role-specialized sub-agents. We have added this observation to Section 4.2.
>
> > What is the typical and maximum number of sub-agents deployed during execution, and how does this relate to the performance advantage claimed for the proposed framework? The paper states that the agent can spawn unlimited sub-agents, but what was the actual peak number observed during the experiments?*
>
> We observed a peak of 8 sub-agents active in parallel, with an average of 2.82 concurrent across runs. This parallelism enabled ARTEMIS to pursue multiple leads and maintain sustained progress over long horizons. We have added this detail to Section 4.2.
>
> > What is the detailed cost breakdown for each major component (such as the supervisor, sub-agents, and triage module) to better understand which parts contribute most to the total operational cost?*
>
> In decreasing order of cost: (1) sub-agents, (2) supervisor, (3) triage module. We tracked total system costs via dedicated API keys but did not log component-level token counts, precluding precise per-module figures. Because sub-agents dominate cost, using more economical models (e.g., o4-mini) would yield the most significant savings. We have added this information to Section 5.4.

---

### Meta-Review · Program_Chairs · 2026-01-06

**Summary:**

This paper presents a large-scale live study comparing autonomous AI agents with professional penetration testers on a real university enterprise network. The reviewers generally agree that this work is novel, impactful, and difficult to achieve, providing rare real-world evidence about agentic systems’ offensive security capabilities, limitations, and cost profiles. The proposed ARTEMIS framework demonstrates strong performance relative to both existing agent systems and human professionals, and the study offers valuable qualitative and quantitative insights that go well beyond synthetic benchmarks.

**This paper is being conditionally accepted provided the authors address the following:**
[Ethics concerns] Reviewers raised ethics concerns that the authors responded to in the rebuttal. The actions that the authors mentioned in their response should be described in detail in the camera ready version.
**Conditions for acceptance have been satisfied**.

**Reviewer Concerns:**

The main outstanding concerns focus on the non-reproducibility of a live enterprise environment, the need for clearer articulation of ARTEMIS’s technical novelty, limited analysis of failure modes, the lack of comparison across different underlying models, and questions about experimental fairness and operational details. The authors’ rebuttal substantially addresses these issues by expanding the description of ARTEMIS’s architecture and planning mechanisms, adding explicit failure-mode analysis, clarifying the time-budget comparison with human participants, and explaining the choice to prioritize state-of-the-art models to estimate upper-bound risk. While reproducibility remains inherently limited due to the nature of the setting, the authors provide a reasonable justification and outline concrete steps toward future replicable environments. Overall, the remaining issues are primarily about scope, reporting depth, and future extensions rather than fundamental soundness.

**Reviewer Scores:**

Given the rebuttal, I expect the more positive reviewers to maintain their high scores. The reviewer who initially scored the paper lower due to concerns about novelty clarity, failure analysis, and experimental framing would likely increase their score modestly, as the authors have directly and substantively responded to these points. Overall, I believe the reviewer scores would converge toward a clearer acceptance range rather than remaining borderline.

---

### Decision · Program_Chairs · 2026-01-26

Accept (Poster)